# MicroRNA-29a Counteracts Glucocorticoid Induction of Bone Loss through Repressing TNFSF13b Modulation of Osteoclastogenesis

**DOI:** 10.3390/ijms20205141

**Published:** 2019-10-17

**Authors:** Re-Wen Wu, Wei-Shiung Lian, Yu-Shan Chen, Chung-Wen Kuo, Huei-Ching Ke, Chin-Kuei Hsieh, Shao-Yu Wang, Jih-Yang Ko, Feng-Sheng Wang

**Affiliations:** 1Department of Orthopedic Surgery, Kaohsiung Chang Gung Memorial Hospital, Kaohsiung 83301, Taiwan; ray4595@gmail.com; 2Core Laboratory for Phenomics and Diagnostic, Kaohisung Chang Gung Memorial Hospital, Kaohsiung 83301, Taiwan; lianws@gmail.com (W.-S.L.); ggyy58720240@gmail.com (Y.-S.C.); bulakuo@gmail.com (C.-W.K.); maggie2258@cgmh.org.tw (H.-C.K.); jorno0329@gmail.com (C.-K.H.);; 3Department of Medical Research, Kaohisung Chang Gung Memorial Hospital, Kaohsiung 83301, Taiwan; 4Graduate Institute of Clinical Medical Sciences, Chang Gung University College of Medicine, Kaohsiung 83301, Taiwan

**Keywords:** miR-29a, osteoclasts, bone loss, TNFSF13b

## Abstract

Glucocorticoid excess escalates osteoclastic resorption, accelerating bone mass loss and microarchitecture damage, which ramps up osteoporosis development. MicroRNA-29a (miR-29a) regulates osteoblast and chondrocyte function; however, the action of miR-29a to osteoclastic activity in the glucocorticoid-induced osteoporotic bone remains elusive. In this study, we showed that transgenic mice overexpressing an miR-29a precursor driven by phosphoglycerate kinase exhibited a minor response to glucocorticoid-mediated bone mineral density loss, cortical bone porosity and overproduction of serum resorption markers C-teleopeptide of type I collagen and tartrate-resistant acid phosphatase 5b levels. miR-29a overexpression compromised trabecular bone erosion and excessive osteoclast number histopathology in glucocorticoid-treated skeletal tissue. Ex vivo, the glucocorticoid-provoked osteoblast formation and osteoclastogenic markers (NFATc1, MMP9, V-ATPase, carbonic anhydrase II and cathepsin K) along with F-actin ring development and pit formation of primary bone-marrow macrophages were downregulated in miR-29a transgenic mice. Mechanistically, tumor necrosis factor superfamily member 13b (TNFSF13b) participated in the glucocorticoid-induced osteoclast formation. miR-29a decreased the suppressor of cytokine signaling 2 (SOCS2) enrichment in the TNFSF13b promoter and downregulated the cytokine production. In vitro, forced miR-29a expression and SOCS2 knockdown attenuated the glucocorticoid-induced TNFSF13b expression in osteoblasts. miR-29a wards off glucocorticoid-mediated excessive bone resorption by repressing the TNFSF13b modulation of osteoclastic activity. This study sheds new light onto the immune-regulatory actions of miR-29a protection against glucocorticoid-mediated osteoporosis.

## 1. Introduction

Glucocorticoid overmedication dysregulates bone cell activity, accelerating bone mass loss and structure deterioration, which increases the development of osteoporotic diseases [1]. Osteoclast overdevelopment that progressively erodes bone microstructure [2] and weakens biomechanical properties of skeletal tissues [3] is a notable pathological feature of glucocorticoid-induced osteoporosis. Increasing studies have shown the involvement of intracellular pathways, like vanilloid receptor, cannabinoid receptor [4] and circadian rhythm [5], in glucocorticoid-mediated osteoclast activation. 

On the other hand, glucocorticoids also facilitate osteogenic cells and bone marrow cells to produce osteoclastogenesis-promoting factors, like tumor necrosis factor superfamily (TNFSF) member receptor activator NF-κB ligand (RANKL), which is indispensable in osteoclast differentiation and activation [6]. Several TNFSF members are also involved in the RANKL-independent osteoclast differentiation program. Of them, CD137 promotes osteoclast migration and differentiation in breast cancer-induced bone metastasis [7]. Mice deficient in TNFSF14 show increased bone mass together with suppressed osteoclastic resorption [8]. TNFSF13b promotes the shifting of human monocytes into osteoclastic cells [9], whereas blocking TNFSF13b reverses multiple myeloma-mediated osteoclast formation [10]. 

MicroRNA belong to non-coding small RNA, disrupting mRNA targets and regulating tissue development, remodeling and malignancy [11]. Many microRNAs alter osteoclast survival, differentiation and maturation under osteoporotic and arthritic conditions. For example, miR-27a mediates the estrogen-induced loss of osteoclast differentiation and bone resorption capacity of bone-marrow macrophages [12]. miR-31a-5p accelerates osteoclast formation in an age-mediated osteoporotic skeleton [13]. miR-124 downregulates osteoclastic resorption in arthritic joints in rats upon adjuvant injection [14]. miR-34 inhibits osteoclast activation in the development of osteoporosis and osteolytic bone metastasis [15]. Moreover, the miR-29 family regulates myogenesis of muscle stem cells [16], chondrogenic differentiation of mesenchymal stem cells [17] and fibrosis matrix formation in inflamed tissue [18]. miR-29 signaling also modulates immune cell activation in multiple sclerosis [19], the host immune response [20] and inflammatory reaction in colitis [21]. We previously revealed that glucocorticoid induced bone loss and marrow adipose overdevelopment along with decreased miR-29a expression [22]. The biological function of miR-29a in osteoclast behavior in glucocorticoid-treated bone tissue is not well understood. 

This study aimed to investigate whether osteoclast differentiation or bone resorption in the glucocorticoid-induced osteoporotic skeleton was changed in miR-29a transgenic mice (miR-29aTg) and tested whether cytokine TNFSF13b mediated the miR-29a regulation of glucocorticoid-provoked osteoclast formation.

## 2. Results

### 2.1. miR-29 Overexpression Compromised Glucocorticoid-Induced Bone Loss

We utilized miR-29aTg mice to test whether increasing miR-29a transcripts in trabecular bone, as evident from in situ hybridization (Figure 1A), altered bone mass or osteoclastic resorption in skeletal tissue upon glucocorticoid treatment. Bone resorption markers C-teleopeptide of type I collagen (CTX-1) and tartrate-resistant acid phosphatase 5b (TRAP5b) levels in sera were significantly increased in wild-type (WT) mice upon 5 mg/kg/day methylprednisolone treatment for 28 days, whereas they were significantly downregulated in glucocorticoid-treated miR-29aTg mice (Tg) (Figure 1B). Of note, glucocorticoid-treated WT mice exhibited severely poor trabecular bone microstructure (Figure 1C) along with significantly decreased bone mineral density and increased cortical bone porosity (Figure 1D). Well-woven trabecular bone architecture together with minor bone loss and porosity occurred in glucocorticoid-treated miR-29aTg mice.

### 2.2. miR-29 Repressed the Glucocorticoid-Induced Osteoclastic Erosion Histopathology

In addition, bone tissue in glucocorticoid-treated WT mice showed severe trabecular loss and increased osteoclast formation histopathology as evident from TRAP (tartrate-resistant acid phosphatase) staining, whereas specimens from glucocorticoid-treated miR-29aTg mice displayed abundant trabecular bone together with mild osteoclast distribution (Figure 2A). Consistently, glucocorticoid significantly increased trabecular separation (Tb.Sp; Figure 2B), osteoclast number (Oc.N; Figure 2C), erosion area (Figure 2D) and eroded surface (ES.BS%; Figure 2E) in WT mice. miR-29a overexpression reversed the bone resorption histomorphology in glucocorticoid-treated skeleton.

### 2.3. miR-29a Inhibited Osteoclast Differentiation and Resorption Capacity

The miR-29a improvement of bone erosion in glucocorticoid-treated bone tissue prompted us to isolate primary bone-marrow macrophages for characterizing osteoclast activity in WT mice and miR-29a mice. Numerous enlarged osteoclasts positive for TRAP staining formed in glucocorticoid-treated WT mice; these phenomena were improved in the glucocorticoid-treated miR-29aTg group (Figure 3A). Glucocorticoid significantly increased osteoclast number and area (Figure 3B) and also upregulated osteoclastogenic markers NFATc1, cathepsin K (Figure 3C), mature osteoclast markers carbonic anhydrase II and vacuolar H^+^-ATPase expression (Figure 3D) in the WT group. miR-29a overexpression significantly downregulated osteoclast formation and osteoclast marker expression of bone-marrow macrophages below the baseline and also improved the glucocorticoid-upregulated osteoclast differentiation.

In addition, osteoclasts in glucocorticoid-treated wild type (WT) mice showed strongly fluorescent F-actin ring morphology (Figure 4A) along with significant increases in F-actin rings (Figure 4B) and matrix metallopeptidase 9 (MMP9) expression (Figure 4C). miR-29a overexpression significantly repressed these reactions in osteoclasts from glucocorticoid-treated skeleton. Moreover, osteoclast precursor cells were incubated onto the bone biomimetic surface to characterize pit formation (Figure 4D). Osteoclasts from glucocorticoid-treated WT mice eroded larger area of pits as compared with vehicle-treated WT mice. This activity was significantly downregulated in glucocorticoid-treated miR-29aTg mice (Figure 4E). Gain of miR-29 signaling significantly reduced osteoclastic resorption capacity below the baseline. 

### 2.4. TNFSF13b-Mediated miR-29a Regulation of Osoteoclast Formation

Bioinformatics reveal that TNFSF13b is one of putative targets of miR-29 (www.microrna.org). This cytokine is shown to promote monocyte differentiation toward osteoclastic lineages [9,10]. Of note, osteoblasts adjacent to trabecular bone in glucocorticoid-treated WT mice exhibited strong TNFSF13b immunostaining, whereas bone cells in glucocorticoid-treated bone tissue displayed weak TNFSF13b immunoreactivity (Figure 5A). Consistent with the histomorphometric analysis (Figure 5B), serum TNFSF13b levels were significantly increased in WT mice upon glucocorticoid treatment. miR-29a overexpression attenuated the glucocorticoid-augmented TNFSF13b levels (Figure 5C).

We tested whether TNFSF13b participated in the miR-29a reduction of osteoclastogenic activities in glucocorticoid-treated skeleton. TNFSF13b antibody or TNFSF13b protein were added to bone-marrow macrophages from WT mice and miR-29aTg mice with glucocorticoid treatment. Of interest, blocking TNFSF13b significantly decreased osteoclast formation of macrophage cultures from glucocorticoid-treated WT mice (Figure 5D). On the contrary, co-incubation with TNFSF13b protein significantly increased osteoclast differentiation of bone-marrow macrophages from glucocorticoid-treated miR-29aTg mice (Figure 5D), which is suggestive of the involvement of TNFSF13b in miR-29a regulation of osteoclast differentiation.

### 2.5. SOCS2 Controlled the miR-29a Inhibition of TNFSF13b Signaling

Bioinformatics searches (www.cbil.upenn/cgi-bin/tess/tess) revealed that SOCS2 is a putative transcription factor for TNFSF13b transcription. miR-29a overexpression significantly attenuated the glucocorticoid-induced increases in SOCS2 levels in bone tissue (Figure 6A). In addition, glucocorticoid significantly increased the SOCS2 enrichment in the TNFSF13b promoter as evident from chromatin immunoprecipitation (ChIP)-PCR analysis (Figure 6B) along with a significant increase in TNFSF13b mRNA transcription (Figure 6C) in WT bone tissue. These effects were significantly compromised in miR-29aTg skeletal tissue. SOCS2 monoclonal antibodies used in ChIP experiments exhibited prominent enrichment in the TNFSF13b proximal promoter region as compared to IgG, which is suggestive of high specificity to the promoter region of interest. 

Consistent with the analysis of the in vivo model, glucocorticoid and miR-29a knockdown significantly decreased miR-29a expression in osteoblast cultures (Figure 7A). Gain of miR-29a signaling significantly attenuated the glucocorticoid-provoked TNFSF13b expression (Figure 7B). Moreover, glucocorticoid significantly increased SOCS2 expression. Loss of SOCS2 function (Figure 7C) downregulated TNFSF13b expression in osteoblast cultures upon glucocorticoid stress (Figure 7D). 

## 3. Discussion

Physiological levels of glucocorticoids are essential to maintain the contact of osteoclastic cells with the mineralized matrix, harmonizing the bone turnover reaction beneficial for bone mass homeostasis [6,23]. Glucocorticoid excess increases osteoclast survival [24] and resorption activity [25], overwhelmingly shattering bone mass and structure. While accumulating evidence has revealed the microRNA signaling regulation of osteoclast differentiation and bone erosion in degenerative and metastatic conditions [26,27], little is known of the role microRNAs may play in osteoclast overgrowth in the development of glucocorticoid-induced osteoporosis. Collective analysis in this study revealed that miR-29a downregulated the glucocorticoid-augmented osteoclastic resorption, fending off osteoporotic skeleton development. Decreased cytokine TNFSF13b signaling in bone microenvironment contributed to the miR-29a repression of osteoclast function. This study offers a new epigenetic insight into how the microRNA pathway controls osteoclast behavior, delaying the development of glucocorticoid-induced bone loss. Robust analysis is also the first indication to highlight the contribution of TNFSF13b signaling to this osteoporotic disorder. 

In this study, treatment with a high dose of methylprednisolone deteriorated bone mass along with significant increases in skeletal porosity and serum bone resorption markers, like TRAP5b and CTX-1. Our investigations are in agreement with other groups’ studies showing that glucocorticoid administration increased serum TRAP5b levels [28] and cortical bone loss [29]. Furthermore, miR-29a overexpression attenuated the glucocorticoid-induced osteoblast dysfunction and marrow adipose deposition [22], which is indicative that miR-29a signaling may alter osteoclast behavior in glucocorticoid-treated bone tissue. These findings reasoned us to employ miR-29aTg mice to understand whether miR-29a affected the osteoclastic resorption reaction in the glucocorticoid excess-stressed bone microenvironment.

Of note, a plethora of glucocorticoid-induced excessive bone resorption signs, like resorption marker overproduction, bone cortical porosity and trabecular bone erosion histopathology, were significantly downregulated in miR-29aTg mice, which is suggestive of miR-29a signaling suppression of bone remodeling in glucocorticoid-treated skeletal tissue. The analysis of decreased osteoclast distribution in miR-29aTg bone tissue underpinned the investigations of its protective effects on bone mass homeostasis. In addition, miR-29a overexpression reversed the glucocorticoid upregulation of osteoclast differentiation capacity, like osteoclast formation, maturation and pit formation. The investigations of osteoclast differentiation of bone-marrow macrophages were consistent with the in vivo findings. However, the effect of the miR-29 family members on osteoclastogenic lineage specification or osteoclast maturation remains uncertain. For example, knocking down miR-29a, b or c decreases osteoclast differentiation capacity of RAW264.7 monocytic cells, whereas survival and F-actin ring formation were not significantly affected in mature osteoclasts [30]. Treatment with miR-29a mimic promotes osteoclast survival rather than differentiation; however, the miR-29a inhibitor downregulates lipopolysaccharide-induced osteoclast growth [31]. On the contrary, forced miR-29b expression inhibits osteoclast markers nuclear factor of activated T-cell, cytoplasmic 1 (NFATc1) and matrix metallopep (MMP9) expression and also reduces collagen degradation and pit formation of osteoclastic cultures in a multiple myeloma-mediated osteolysis model [32]. This study uncovered that miR-29a overexpression suppressed osteoclast formation in glucocorticoid excess-treated skeleton. We speculated that the miR-29 action to osteoclast formation may depend on osteoporotic disease and osteoclastogenic progenitor cell types. The repressed osteoclastic resorption in glucocorticoid-treated miR-29aTg bone tissue further explains the complex nature of glucocorticoid-induced osteoporosis. 

The miR-29a hinderance of osteoclast activity promoted us to pinpoint what cytokine contributed to this reaction. TNFSF members, like RANKL, TNFSF13b and TNFSF14, etc., regulate differentiation, maturation and activation of osteoclastogenic cells in the development of osteoporosis caused by estrogen loss, arthritis or bone metastasis [33,34]. A decrease in RANKL rather than osteoprotegerin (OPG) expression occurs in the miR-29a regulation of glucocorticoid excess-treated skeletons [22]. TNFSF13b is shown to promote osteoclast differentiation of human monocytic cultures in the absence of RANKL [9]. Of note, this study uncovered the involvement of TNFSF13b in the glucocorticoid-provoked osteoclast activities, as significant increases in TNFSF13b secretion and immunostaining occurred in glucocorticoid-treated mice. TNFSF13b antibody blockade mitigated the glucocorticoid upregulation of osteoclast formation of bone-marrow macrophages. Adding TNFSF13b protein to miR-29aTg bone-marrow macrophages weakened the miR-29a inhibition of osteoclast formation. Intriguing analysis sheds a new light on the TNFSF action in the development of glucocorticoid-induced bone damage. 

Furthermore, SOCS2 is a master transcription factor regulating polarization and lineage commitment of macrophages [35]. Mice deficient in SOCS2 show increased bone growth and widened growth plate [36] and display an amplified response to growth hormone promotion of bone development [37]. This immune regulator is a putative transcription factor for TNFSF13b transcription (www.cbil.upenn/cgi-bin/tess/tess). Collective analysis confirmed that miR-29a overexpression attenuated the glucocorticoid upregulation of SOCS2 levels and the SOCS2 occupancy in TNFSF13b promoter, which increased TNFSF13b expression. In vitro, knockdown of SOCS2 downregulated the glucocorticoid-mediated TNFSF13b expression in osteoblasts. The miR-29 family members are emerging immune regulators for immune disorders [18,19], atherosclerotic diseases [38], and malignant tumor formation [39,40]. This study reveals a new immune-regulatory mechanistic by which miR-29a delays osteoclastic resorption in glucocorticoid-treated bone tissue. We do not exclude the possibility that other TNFSF members may be involved in the miR-29a protection from glucocorticoid aggravation of osteoclastic resorption and bone loss. Serine/threonine kinase Pim together with NFκB pathways is shown to regulate TNFSF13b-mediated survival of multiple myeloma cells and osteoclasts [41]. The molecular events underlying TNFSF13b modulation of glucocorticoid-mediated osteoclast formation and the effect of a TNFSF13b inhibitor, like Belimumab, on the glucocorticoid-induced excessive osteoclastic resorption warrants investigation in the future. 

Taken together, miR-29 signaling represses the glucocorticoid-induced excessive osteoclast formation and bone resorption, slowing the development of osteoporotic skeleton. miR-29a improves the glucocorticoid-induced osteoclast overdevelopment by reducing SOCS2 and TNFSF13b signaling (Figure 8). Profound analyses convey a new epigenetic insight into microRNA shielding from glucocorticoid-induced excessive bone remodeling and bone loss.

## 4. Materials and Methods

### 4.1. miR-29a Transgenic Mice

Protocols for animal breeding, experimentation and care were reviewed and approved by the Institutional Animal Use and Care Committee in March, 2011, Kaohsiung Chang Gung Memorial Hospital in September, 2012 (Affidavit No. 2011030701 and No.2012091003). (Friend leukemia virus B; FVB) mice overexpressing miR-29a precursor (FVB/TNar-Tg-29a/PGK; miR-29aTg) driven by the phosphoglycerate kinase (PGK) promoter were bred, as previously described [22]. The genotype of each animal was confirmed using customized primers (forward: 5′-GAGGATCCCCTCAAGGATACCAAGGGATGAAT-3′; reverse, 5′-CTTCTAGAAGGAGTGTTTCTAGGTATCCGTCA-3′) along with PCR analysis [22]. 

### 4.2. Glucocorticoid-Induced Osteoporosis

Male wild-type mice (WT) and miR-29aTg mice (Tg; 12 weeks old) were intraperitoneally given 5 mg/kg/day methylprednisolone or vehicle for 28 days. Upon euthanasia, peripheral blood was harvested via an intra-cardiac puncture, and tibia and femurs were dissected for μCT and histological assessment. 

### 4.3. Quantification of Serum Bone Resorption Markers 

Designated ELISA kits were utilized to quantify tartrate-resistant acid phosphate 5b (TRAP5b; Biomedical Technologies Inc., Stoughton, MA, USA), C-teleopeptide of type I collagen (CTX-I; Nordic Bioscience Diagnostics, Herlev, Denmark) and TNFSF13b (R & D Systems, Minneapolis, MN, USA) levels in sera, according to the manufacturers’ manuals. 

### 4.4. Assay of Bone Mass and Microstructure

μCT analysis of bone mineral density, trabecular microstructure and cortical bone porosity was performed using a Skyscan 1176 μCT system (Bruker, Kontich, Belgium). Specimens were subjected to 50-kev, 500-μA, and 69-ms radiography and followed by reconstructing 200 slices of radiographs (isotropic 9-μm voxel each slice) into 3D images using SKYSCAN^®^ CT-Analyzer software (Bruker, Kontich, Belgium). Trabecular bone mineral density (mg/cm^3^) and cortical bone porosity (%) were quantified according to the manufacturer’s instructions. 

### 4.5. In Situ Hybridization and Histomorphometry

Customized miR-29a probes conjugated with digoxigenin (Applied Biosystems, Carlsbad, CA, USA) along with IsHyb In Situ Hybridization kits (Biochain Institute, Inc., Newark, CA, USA) were utilized to probe miR-29a transcripts in bone tissue, as previously described [22]. Specimens were subjected to hematoxylin and eosin staining and tartrate-resistant acid phosphatase histochemical staining. Osteoclast morphology was microscopically evaluated, as previously described [42]. Three fields in each section and 12 sections from 6 mice were randomly selected for histomorphometry using a Ziess microscope (ZIESS, Munchen, Germany) and Image-Pro^®^ Plus image-analysis software (Media Cybernetics Inc., Rockville, MD, USA). Trabecular separation (%), osteoclast number (Oc.No/mm), erosion area (μm^2^), and erosion surface (ES.BS%) were calculated [43].

### 4.6. Ex Vivo Osteoclast Differentiation and F-Actin Ring Immunofluorescence Labeling

Primary bone-marrow macrophages in tibiae and femurs were isolated, as previously described [44]. In brief, nucleated cells in bone marrow were isolated upon lysis of red blood cells using RBC Lysis buffer (Sigma-Aldrich Co., St Louis, MO, USA) and incubated in α-MEM with 10% fetal bovine serum and 20 ng/mL M-CSF (R&D Systems, Minneapolis, MN, USA) for 1 day. The floating cells were harvested upon incubation. Then, 10^5^/well macrophages (24-well plates) were incubated in osteoclastogenic medium containing α-MEM, 10% fetal bovine serum, 20 ng/mL M-CSF and 20 ng/mL RANKL (R&D Systems, Minneapolis, MN, USA) for 7 days. In a subset of the experiment, 10^5^/well macrophages were incubated in a mixture of osteoclastogenic medium containing 20 ng/mL TNFSF13b monoclonal antibody, IgG or recombinant TNFSF13b (R&D Systems, Minneapolis, MN, USA). After incubation, cell cultures were subjected to TRAP cytochemical staining (Sigma-Aldrich Co., St Louis, MO, USA). F-actin ring formation in osteoclasts was probed using F-actin antibody conjugated with Alexa Fluor^®^ 488 Phalloidin (Life Technologies, Grand Island, NY, USA) and DAPI-Fluoromount G (Southern Biotech, Birmingham, AL, USA). The number and area of multinuclear cells positive for TRAP stain and fluorescence F-actin rings in 3 fields of each well and 18 wells from 6 animals were counted using a Zeiss inverted microscope and image-analysis software. 

### 4.7. Pit Formation

In total, 105 primary bone-marrow macrophages were seeded onto an Osteo Assay Stripwell Plate (Corning, Lowell, MA, USA), which was coated with a bone biomimetic synthetic surface, followed by incubating in osteoclastogenic medium for 7 days, according to the manufacturer’s instructions. Culture wells in the absence of primary cells were used as a blank control group. After removal of cell cultures, each well was rinsed with deionized water and then stained by von Kossa staining. Pits that were negative for von Kossa staining on the bone biomimetic synthetic surface were subjected to microscopy [45]. Areas of pits in 3 fields of each well and 18 wells from 6 mice were counted. Percentile pit formation was calculated as areas of pits/areas of microscopic field × 100%.

### 4.8. Quantitative RT-PCR 

Total RNA in macrophage cultures and bone tissue was isolated using QIAzol reagent (Qiagene, Valencia, CA, USA). Upon reverse transcription of 1 μg total RNA, mRNA expression was detected using primers for NFATc1 (forward: 5′-GAAGGTGTACTCCTCGGGTGG-3′; reverse: 5′-GATACCTGGCTCGGTAACACCAC-3′), V-ATPase (forward: 5′-AGAAAGCCAAGTGCCTACTCC-3′; reverse: 5′-AAAGGGAAGGGTTTCTTTTGG-3′), MMP9 (forward: 5′-GGGAAGGCTCTGCTGTTCA-3′; reverse: 5′-CGGTTGAAGCAAAGAAGGAG-3′), cathepsin K (forward: 5′-CCTGCGGCATTACCAACAT-3′; reverse: 5′-GCTGCAGGACTCCAATGTCT-3′), TNFSF13b (forward: 5′-TTCCATGGCTTCTCAGCTTT-3′; reverse: 5′-CGTCCCCAAAGACGTGACT-3′), and actin (forward: 5′-GACGGCCAGGTCATCACTAT-3′; reverse: 5′-CTTCTGCATCCTGTCAGCAA-3′). Equation 2^−ΔΔCt^, where ΔΔCt = ΔΔCt_glucocorticoid_ − ΔCt_vehicle_ and ΔΔCt = Ct_gene_ − Ct_actin_ was adopted to calculated the relative expression of each gene. 

### 4.9. Immunoblotting

Protein lysates in bone tissue was extracted using PRO-PREP™ Extraction Kits (iNtRON Biotechnology, Sungnam, Korea), according to the manufacturer’s instruction. SOCS2 and actin in bone tissue proteins were probed using Western blotting protocols along with SOCS2 and actin antibodies (Cell Signaling Technology, Danvers, MA, USA) and horseradish peroxidase-conjugated IgG that was visualized by chemiluminescence agents.

### 4.10. Chromatin Immuneprecipitation (ChIP)-PCR

Bone tissue extracts were immunoprecipitated with SOCS2 monoclonal antibodies or IgG (Millipore, Billerica, MA, USA). For isolating chromatin, the immunocomplexes were subjected to sonication, elution and Proteinase K digestion using Megan ChIP A/G kits (Millipore, Billerica, MA, USA), according to the manufacturer’s instructions. Chromatin was probed by Cy3-labeled primers (forward: 5′-CGTCCTTTGGTCTTGCACTT-3′; reverse: 5′-GGATTGTGGGTTCAG GGTTA-3′) for TNFSF13b proximal promoter region (NCBI Accession: NM_033622) using PCR protocols. Enrichment of SOCS2 in designated promoter regions was calculated as % input as previously described [46].

### 4.11. Transfection

For transfection, 1 nM miR-29a precursor, 1 nM miR-29a antisense oligonucleotide or 1 nM scramble control (Applied Biosystems-Ambio Inc., Austin, TX, USA) were mixed with Lipofectamine 2000 (Invitrogen; Thermo Fisher Scientific, Inc., Waltham, MA, USA), according to the manufacturer’s instructions. MC3T3-E1 osteoblasts (5 × 10^5^ cells/well, 6-well plates) were transfected with the mixtures followed by incubating in DMEM with 10% fetal bovine serum with or without 1 μM dexamethasone for 24 h, as previously described [22]. In a subset experiment, cell cultures were transfected with 1 μg SOCS2 RNAi and incubated in 1 μM dexamethasone for 24 h. Total RNA in cell cultures was isolated for RT-qPCR analysis of miR-29a, SOCS2 and TNFSF13b expression. 

### 4.12. Statistical Analysis

Data were expressed as means ± standard errors. Differences among miR-29a transgenic mice and wild-type mice with glucocorticoid or vehicle treatment were analyzed by a parametric ANOVA test and a Bonferroni post-hoc test. A *p* value of <0.05 was considered statistically significant.

## Figures and Tables

**Figure 1 ijms-20-05141-f001:**
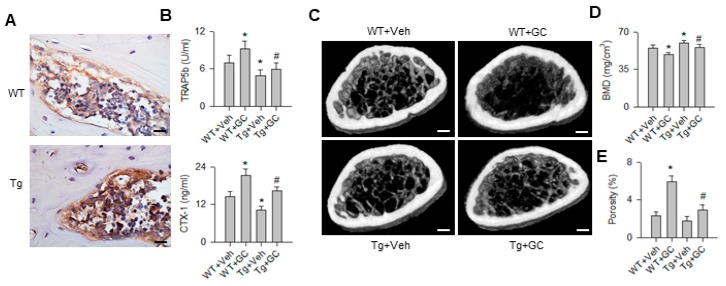
Analysis of bone mass, microstructure and resorption markers in bone tissue in WT mice and miR-29aTg mice. Strong miR-29a transcripts in miR-29aTg bone tissue (**A**); scale bar, 10 μm. miR-29a overexpression downregulated serum TRAP5b and CTX-1 levels (**B**). Glucocorticoid-treated WT mice showed sparse trabecular bone, whereas abundant trabecular microstructure remained in glucocorticoid-treated miR-29aTg mice (**C**); scale bar, 5 mm. miR-29a overexpression improved bone mineral density (**D**) and cortical bone porosity (**E**) in glucocorticoid-treated skeleton. Data are expressed as the mean ± standard errors calculated from 6 mice. Asterisks * indicate significant differences from the WT + Veh group and hashtags # indicate significant differences from the WT + GC group (*p* < 0.05). WT, wild-type mice; Tg, miR-29aTg mice; Veh, vehicle; GC, glucocorticoid. TRAP5b, tartrate-resistant acid phosphatase 5b; CTX-1, C-telopeptide of type I collagen; BMD, bone mineral density.

**Figure 2 ijms-20-05141-f002:**
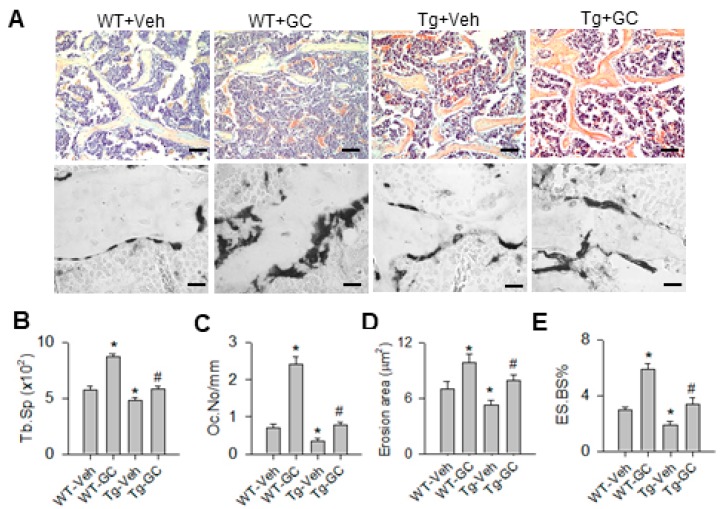
Histological analysis of trabecular bone and osteoclast distribution. Severe trabecular bone loss and increased TRAP-stained osteoclasts existed in glucocorticoid-treated WT bone tissue, whereas well-connected bone histology but few osteoclasts remained in glucocorticoid-treated miR-29aTg bone tissue (**A**); Scale bar, 30 μm (upper panels); 10 μm (lower panel). The glucocorticoid-mediated increases in Tb.Sp (**B**), Oc.N (**C**), erosion area (**D**) and ES.BS% (**E**) were significantly improved in miR-29aTg mice. Data are expressed as the mean ± standard errors calculated from 6 mice. Asterisks * indicate significant differences from the WT-Veh group and hashtags # indicate significant differences from the WT-GC group (*p* < 0.05). WT, wild-type mice; Tg, miR-29aTg mice; Veh, vehicle; GC, glucocorticoid; Tb.Sp, trabecular separation; Oc.N, osteoclast number; ES.BS, eroded surface.

**Figure 3 ijms-20-05141-f003:**
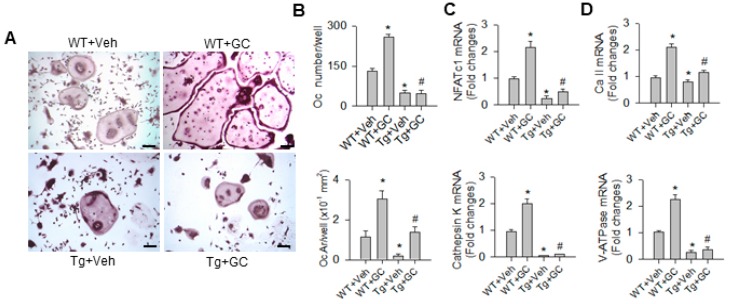
Analysis of osteoclast differentiation of primary bone-marrow macrophages. Increased and enlarged osteoclasts positive for TRAP staining occurred in glucocorticoid-treated WT mice, whereas few osteoclasts formed in miR-29aTg mice (**A**) scale bar, 8 μm. miR-29a overexpression repressed the glucocorticoid-induced increases in osteoclast number and area (**B**) and also reduced osteoclastogenic markers NFATc1, cathepsin K (**C**), and osteoclast maturation markers carbonic anhydrase II and V-ATPase expression (**D**). Data are expressed as the mean ± standard errors calculated from 6 mice. Asterisks * indicate significant differences from the WT + Veh group and hashtags # indicate significant differences from the WT + Veh group (*p* < 0.05). WT, wild-type mice; Tg, miR-29aTg mice; Veh, vehicle; GC, glucocorticoid, Oc, osteoclasts; Oc.Ar, osteoclast area; NFATc1, nuclear factor of activated T-cells-c1; Ca II, carbonic anhydrase II; V-ATPase, vacuolar H^+^-ATPase.

**Figure 4 ijms-20-05141-f004:**
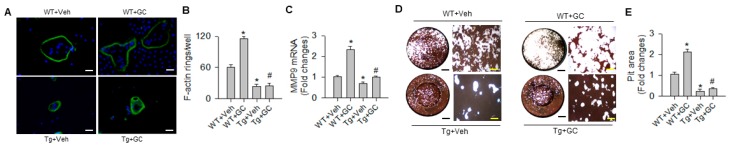
Analysis of F-actin ring formation and pit formation of bone-marrow osteoclastogenic cells. Osteoclasts in glucocorticoid-treated WT mice showed strongly fluorescent F-actin ring morphology (**A**) (scale bar, 20 μm) and increases in F-actin ring number (**B**), MMP9 expression (**C**) and pit formation (**D**, **E**); (black scale bar, 7 mm; yellow scale bar, 30 μm). These effects were compromised in glucocorticoid-treated miR-29aTg mice. Data are expressed as the mean ± standard errors calculated from 6 mice. Asterisks * indicate significant differences from the WT + Veh group and hashtags # indicate significant differences from the WT + GC group (*p* < 0.05). WT, wild-type mice; Tg, miR-29aTg mice; Veh, vehicle; GC, glucocorticoid; MMP9, matrix metalloproteinase 9.

**Figure 5 ijms-20-05141-f005:**
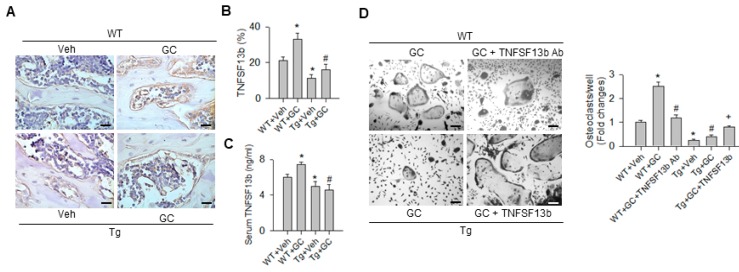
Analysis of TNFSF13b immunohistochemistry and action in osteoclast formation. Bone cells in glucocorticoid-treated WT mice showed strong TNFSF13b immunostaining, but expressed weak immunoreaction in glucocorticoid-treated miR-29aTg mice (**A**); scale bar, 10 μm. miR-29a overexpression attenuated the glucocorticoid-induced increases in TNFSF13b immunostaining (**B**) and serum TNFSF13b levels (**C**). TNFSF13b antibody blockade attenuated the glucocorticoid-induced osteoclast formation of bone-marrow macrophages from WT mice, whereas TNFSF13b protein increased osteoclast formation of bone-marrow macrophages from glucocorticoid-treated miR-29aTg mice (**D**); scale bar, 8 μm. Data are expressed as the mean ± standard errors calculated from 6 mice. Asterisks * indicate significant differences from the WT + Veh group, hashtags # indicate significant differences from the WT + GC group and plus + indicate significant difference the Tg + GC group (*p* < 0.05). WT, wild-type mice; Tg, miR-29aTg mice; Veh, vehicle; GC, glucocorticoid; TNFSF13b, tumor necrosis factor superfamily 13b; Ab, antibody.

**Figure 6 ijms-20-05141-f006:**
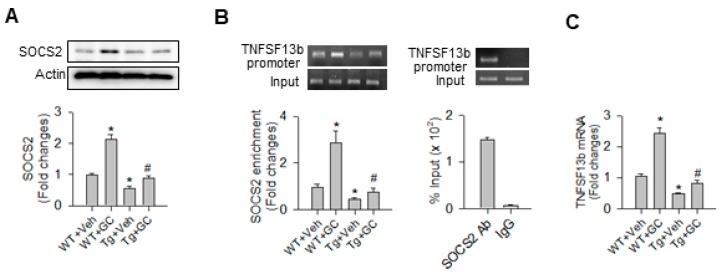
Analysis of SOCS2 immunoblotting and ChIP-PCR of SOCS2 enrichment in TNFSF13b. miR-29a overexpression attenuated the glucocorticoid-induced increase in SOCS2 protein (**A**), the SOCS2 enrichment in the TNFSF13b promoter region (**B**), and TNFSF13b mRNA expression (**C**). Data are expressed as the mean ± standard errors calculated from 3–6 mice. Asterisks * indicate significant differences from the WT + Veh group and hashtags # indicate significant differences from the WT + GC group (*p* < 0.05). Veh, vehicle; GC, glucocorticoid; SOCS2, suppressor of cytokine signaling 2; Ab, antibody.

**Figure 7 ijms-20-05141-f007:**
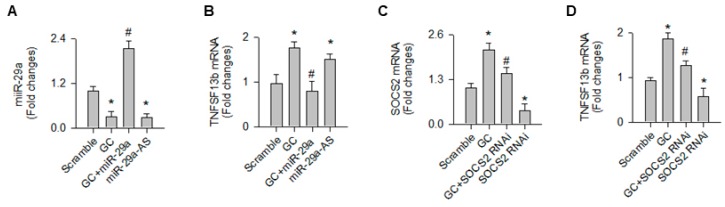
Analysis of TNFSF13b and SOCS2 expression in osteoblasts. Forced miR-29a expression (**A**) attenuated the glucocorticoid-provoked TNFSF13b expression in osteoblasts, whereas loss of miR-29a function increased the cytokine expression (**B**). SOCS2 knockdown (**C**) attenuated the glucocorticoid-mediated increases in TNFSF13b expression (**D**). Data are expressed as the mean ± standard errors calculated from 3 experiments. Asterisks * indicate significant differences from the scramble group and hashtags # indicate significant differences from the GC group. GC, glucocorticoid; miR-29a-AS, miR-29a antisense oligonucleotide; RNAi, RNA interference.

**Figure 8 ijms-20-05141-f008:**
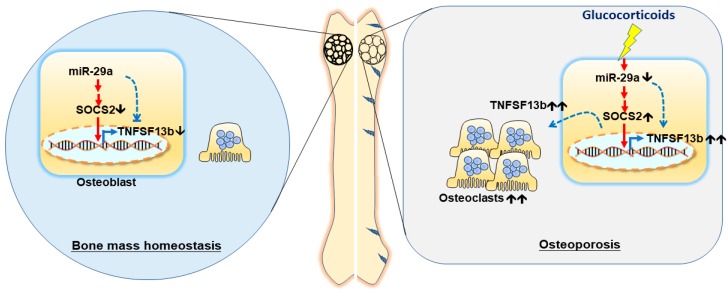
miR-29a attenuates the glucocorticoid-induced TNFSF13b overproduction, improving excessive osteoclast formation in the development of glucocorticoid-mediated osteoporosis.🡹, upregulation; 🡻, downregulation; and 🡹🡹, overexpression. Blue dotted line, a direct action; red solid line, an indirect effect.

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
