# Peer review of "MicroRNA-29a Counteracts Glucocorticoid Induction of Bone Loss through Repressing TNFSF13b Modulation of Osteoclastogenesis"

_ijms, 2019, doi:10.3390/ijms20205141_

Round 1
Reviewer 1 Report
All issues raised in the previous review have been addressed.
Author Response
We are grateful for the approval of the improvement of our revised study by Reviewer 1.
Reviewer 2 Report
The issues raised have been well addressed and I have no further comments.
Author Response
We are grateful for the approval of the improvement of our revised study by Reviewer 2.
Reviewer 3 Report
No additional data has been shown in the revised manuscript which does not satisfy the reviewer. Just describing the limitation is not enough for the revision in this journal.
Author Response
Response to Reviewer #3
No additional data has been shown in the revised manuscript which does not satisfy the reviewer. Just describing the limitation is not enough for the revision in this journal.
Authors’ response: Thank you for this comment. This study provided additional evidence that gain of miR-29a expression attenuated the glucocorticoid-provoked TNFSF13b expression in osteoblasts. miR-29a knockdown increased the cytokine expression. Moreover, SOCS2 knockdown downregulated the glucocorticoid-mediated TNFSF13b expression in osteoblasts. We re-wrote the sentences in the 2nd revised version (Line 194 – 197 and Fig 7), which now read as follows:
Consistent with the analysis of in vivo model, glucocorticoid and miR-29a knockdown significantly decreased miR-29a expression in osteoblast cultures (Fig.7A). Gain of miR-29a signaling significantly attenuated the glucocorticoid-provoked TNFSF13b expression (Fig. 7B). Moreover, glucocorticoid significantly increased SOCS2 expression. Loss of SOCS2 function (Fig. 7C) downregulated TNFSF13b expression in osteoblast cultures upon glucocorticoid stress (Fig. 7D)
We also re-wrote the experimental protocols of in vitro models in the 2nd revised version (Lines 365-370), which now read as follows:
1 nM miR-29a precursor, 1 nM miR-29a antisense oligonucleotide or 1 nM scramble control (Applied Biosystems-Ambio Inc.) were mixed with Lipofectamine 2000 (Invitrogen), according to the manufacturer’s instructions. MC3T3-E1 osteoblasts (5 ×105 cells/well, 6-well plates) were transfected with the mixtures and followed by incubating in DMEM with 10% fetal bovine serum with or without 1 μM dexamethasone for 24 hours, as previously described [22]. In a subset experiment, cell cultures were transfected with 1 μg SOCS2 RNAi and incubated in 1 μM dexamethasone for 24 hours.Total RNA in cell cultures were isolated for RT-qPCR analysis of miR-29a, SOCS2 and TNFSF13b expression.
Please see the attachment to find revised manuscript.

Round 2
Reviewer 3 Report
I agree with the acceptance of this manuscript.
This manuscript is a resubmission of an earlier submission. The following is a list of the peer review reports and author responses from that submission.
Round 1
Reviewer 1 Report
The authors have shown that osteoclastogenic makers along with F-actin ring development and pit formation of primary bone-marrow macrophages were downregulated in miR-29a transgenic mice. They show that tumor necrosis factor superfamily member 13b (TNFSF13b) participated in the glucocorticoid-induced osteoclast formation. miR-29a decreased the SOCS2 enrichment in TNFSF13b promoter and downregulated the cytokine production.
There are however several issues that need to be addressed, before this manuscript is acceptable for IJMS:
If TNFSF13b is critical for GCs-mediated osteoporosis, BAFF inhibitor belimumab should show the similar effects on bone with mir-29a overexpression. This should be investigated both in vivo and in vitro.
The association between SOCS2 and TNFSF13b is not convincing. Elevated expression of TNFSF13b by GCs is rescued by knocking down SOCS2? This should be examined.
How does TNFSF13b control actin formation and NFATc1, the master TF for osteoclastogenesis? The authors should show some data.
Reviewer 2 Report
Overall:
The study investigates the immune-regulatory role of the microRNA-29a (miR-29a) in glucocorticoid-mediated osteoporosis employing both in vivo and in vitro models. The manuscript is generally well written with clear objectives and hypotheses along each section. The conclusion regarding the suppression role of miR-29a in osteoclastogenesis is supported by the evidence that are presented in a logical order to convey the main message.
Suggestions:
The first 3 paragraphs of the discussion session may require some proof reading to enhance the readability.
It is suggested that the author would add a graphical summary of the proposed pathway through which the miR-29a is proposed to interfere with the most commonly accepted pathway of glucocorticoid-induced osteoclastogenesis.
Minor problems
Figure 2: The saturation of the TRAP staining in Tg+Veh and Tg+GC group looks much higher compared to the WT groups. Please unify the exposure.
Figure 5D: The quality of GC+TNGSF13b image is a little bit out of focus. It would be better to replace it is possible.
Line 88, extra “in”
Line 186, a typo of “harmonizinga”
Line 191, please rephrase “action inhibitory”
Line 195, “dragging the progress” may need rephrase
Line 197, resulted in a low bone mass… may need rephrase.
Line 283, the method presented in reference [43] of another group was too brief. It would be helpful for the researcher to state briefly the macrophage isolation protocol they implemented, and how they confirmed cell types.
Reviewer 3 Report
Good science. All experiments nicely conducted and data clearly presented to support conclusion. Just a few minor things:
The indicators (asterisks, hashtags) for statistic significance in the charts are confusing. Please clarify it. In bone-marrow osteoclastogenic cells from Tg mice, all markers are shown at much lower levels (e.g. Fig 3 B, C and D; Fig 4 B, C; etc.). Please address this issue.Reviewer 4 Report
Int. J Medical Science: ijms-574279-
Title: MicroRNA-29a Counteracts Glucocorticoid Induction of Bone Loss through Repressing TNFSF13b Modulation of Osteoclastogenesis
The present study investigated whether osteoclast differentiation or bone resorption in glucocorticoid-induced osteoporotic skeleton is changed in miR-29a transgenic mice (miR-29aTg) and it was tested whether cytokine TNFSF13b mediated the miR-29a regulation of glucocorticoid-provoked osteoclast formation. The authors could show that miR-29a wards off glucocorticoid-mediated excessive bone resorption through repressing TNFSF13b modulation of osteoclastic activity.
Congratulations to the authors to sheds new light onto the immune-regulatory actions of miR-29a protective from glucocorticoid-mediated osteoporosis. I read the study with great interest. The study is correctly designed, the data and the results are significant and well presented. The materials and methods used are described in great detail. The paper is well written.
Some comments:
In general:
In the figure legends, the statistical information given is not clear for me – please revise/explain.
In the figure legends, please spell out abbreviations and add the abbreviation used in brackets like in line 118 cabonic anhydrase (Ca II) as done for Veh, vehicle; GC, glucocorticoid.
in detail:
Line 60-61 – in the sentence, the verb is missing.
Line 88 - check spelling tissuein = tissue in?
Line 118 – what is meant with V-ATPase?
Line 91 In the text, please explain abbreviation at first use: Tb.SP; Oc.N; ES.BS